# The Neglected Solutions: Local Farming Systems for Sustainable Development in the Amazon

**Gabriel da Silva Medina [1],\* and Claudio Wilson Soares Barbosa [2]**

[1] Faculty of Agronomy and Veterinary Medicine, University of Brasilia, Brasilia 70910-900, Brazil
[2] Department of Rural Development, Serviço Cerne, Altamira 68371-902, Brazil
\* Correspondence: gabriel.medina@unb.br

**Abstract:** The productive inclusion of local communities is one of the main challenges to sustainable rural development in the Amazon. Existing development initiatives often prioritize projects with exogenous production systems; thus, local systems are overlooked, despite their large coverage. Based on surveys conducted in 107 riparian communities and detailed case studies in eight communities doing ranching, logging, and fishing, this study describes local management systems developed by rural communities in the confluence between the Amazon and Xingu Rivers. The study showed that (1) local management systems for buffalo ranching, logging, and fishing agreements were found in 61%, 60%, and 21% of the 107 riparian communities, respectively; (2) these systems are based on local know-how and on technological solutions that are locally available; and (3) the improvement and consolidation of these local systems require governmental support. The study reveals that local and traditional farming practices may underpin sustainable development in the Amazon.

**Keywords:** endogenous development; farmer-driven innovations; local governance; legitimization of marginalized knowledge





## 1. Introduction

Rural development can roughly be divided into two trajectories: (1) productivism, and (2) post-productivism rural development [1]. Initial efforts to transfer technologies to local stakeholders that marked productivism were, over time, replaced by the search for endogenous development initiatives based on local and traditional practices [2]. Farmer-driven innovations became highly appreciated in these post-productivism systems [3].

Developing farming practices adequate to local conditions is essential for the Amazon, a threatened biome that requires solutions for sustainable development and social inclusion [4,5]. One of the main challenges faced by rural populations that live in the region is the sustainable generation of income based on natural resources and existing know-how [6,7].

Efforts to promote sustainable development in the Amazon have prioritized pilot projects focused on exogenous production systems. Examples include projects promoting community forest management based on formal management plans and payment for environmental services, which have shown little economic viability [8,9]. In the 2020s, this externally driven approach was revisited in eco-business projects focused on processing bio-economy products, such as oils and cosmetics, to be sold in market niches abroad [10]. Initiatives of the Brazilian government for sustainable development focus on the creation of conservation units and law enforcement to reduce illegal deforestation [11,12].

These top-down approaches are based on the assumptions of (1) nonexistence or inadequacy of local farming systems, which implies the need for promoting exogenous production systems; (2) communities' incapacity to develop sustainable production systems based on traditional and local knowledge, which results in the promotion of modern technological solutions; and (3) legitimacy of government, NGOs, and the international

community to direct and control communities' practices, leading to alien norms and surveillance by external stakeholders [13]. As a consequence, these top-down initiatives neglect the possibility of a bottom-up sustainable rural paradigm based on local potential [14].

Exogenous initiatives promoted by the Brazilian government or NGOs have support from international cooperation and often more visibility than local traditional practices adopted by communities. For example, one of the most comprehensive efforts for listing ongoing sustainable development initiatives in the Brazilian Amazon identified 148 cases, with only 6% of them reporting promising farming systems, such as agroforestry intensification, crop-forest-livestock integration, and crop-beekeeping-fallow restoration, with none of them being local traditional practices [15].

Many municipalities in the Amazon have agriculture or extractive production as key economic sectors, and family farmers and indigenous and non-indigenous traditional communities represent the essence of these local societies. Thus, when the social fabric of these municipalities is made of family farmers and traditional communities, there is great potential for planning development based on locally established production systems [16,17]. The consolidation of small farmers can generate stable landscape mosaics [6,18], and decrease land grabbing and deforestation by large farmers [19].

The Amazon needs a clear and long-term strategic development project based on existing production systems that build on local realities and capacities. This study describes farming systems developed by local populations for the sustainable development of the Amazon. It aims to (a) survey the coverage of local production systems developed by local communities; (b) describe the local practices that can underpin sustainable management systems; and (c) discuss the role of governments, NGOs, and international donors can play in the support of local traditional communities, promoting rural development in the Amazon.

## 2. Theoretical Framework

Classic research on social sciences has unveiled the traditional way of life of populations in the Amazon and highlighted their most representative production systems [20,21]. Examples of these cases are landscape management by indigenous peoples, management of Brazil nut trees by maroon populations (maroons are descendants of Africans in the Americas who formed settlements away from slavery), and management of forest products and fish natural stocks by riparian communities [22,23]. Despite the value of these studies, there is no current systematic effort to promote the development of these local production systems.

Local production and management systems are the most promising alternatives for regional development, considering: (a) the capacity of local groups to strategically manage their resources; (b) the assumption that practices and agreements developed by communities can be more effective than state management; and (c) the importance of the federal and local government's support to legitimate local systems, mainly when local communities are facing external threats [24].

Communities in the Amazon agricultural frontier face challenges at three levels to successfully establish local systems according to their interests. First, they need to develop management practices; second, they need to organize themselves to enforce their regulations and express themselves politically; and third, communities need to be able to interact with external institutions to have their systems acknowledged by society [25,26]. The recognition of local production systems is dependent on support from external stakeholders such as NGOs and the government [7].

Previous research has shown ways in which forest users can devise rules that regulate harvesting patterns to ensure the sustainability of forest resources over time [24,27,28]. Communities might face restrictions in developing these local farming systems and management norms, including their material and physical constraints [29], as well as eventual power inequalities within communities [30], but successful initiatives have been reported by scientific studies [31,32]. Examples of local management systems include communal ar-

eas for collecting non-timber forest products, regulations over timber extraction by loggers, and fishing agreements regulating commercial fishing by external players [25].

Even when local production systems are established, communities need to be able to interact with external institutions to have their systems acknowledged by society [33], which depends on their ability to liaise with the outside world [34]. A successful meso-level collective action depends on the support of actors with whom communities can interact, and the existence of structures through which groups can communicate [35,36].

Besides communities, other agents such as the government, NGOs, and the private sector also may have a role in fostering the development of local forest governance systems. Studies have highlighted the potential of multi-stakeholder forums in assisting with territorial planning [37] as well as the role of agents from the civil society, state, and market in multi-partner governance systems for forest management [38].

## 3. Materials and Methods

This research was based on case studies conducted in riparian communities in the municipality of Porto de Moz, Brazil, in the confluence between the Amazon and Xingu Rivers. This region was chosen because it is inhabited by riparian populations of family farmers for more than 100 years, which depend on ranching, forest extractivism, fishing, agriculture, and other farming activities for their livelihoods. Aiming to stop land grabbing, illegal logging, and deforestation, and to safeguard the local land tenure rights, local communities managed to create the Verde Para Sempre (Forever Green) Extrativist Reserve (Resex) in the year 2004. This accomplishment required the mobilization of many community associations, the Farmer's Union, and the Committee for the Sustainable Development of Porto de Moz (CSD) and counted on support from different external stakeholders such as NGOs [25]. This process resulted in the mapping of the 107 rural communities that existed in Porto de Moz by 2004 [39].

This study began with a survey of all the 107 rural communities and villages of Porto de Moz, 89 of which are part of the Verde para Sempre Resex and 18 are outside the Resex. At least one key informant (the representative of the communities' association) in each one of the 107 communities was consulted to list the farming practices developed by the members of the communities (Figure 1). Each rural community in Porto de Moz is traditionally represented by a formal association of families. Together, these associations conform the municipal Farmers' Union and also the CDS, which are the largest grass-roots organizations in Porto de Moz.

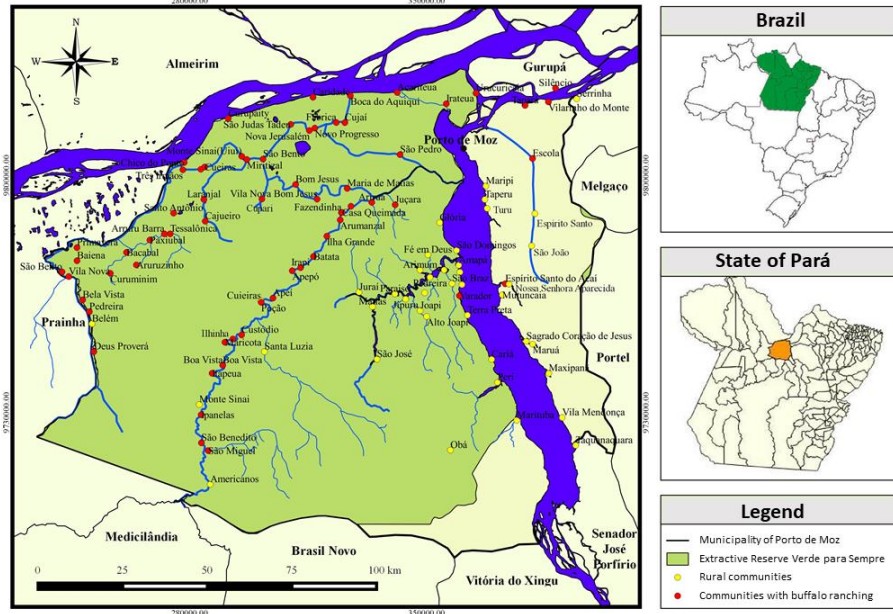

**Figure 1.** Communities with buffalo ranchers in the municipality of Porto de Moz. Source: Fieldwork.

The survey with the representatives of the rural communities was conducted during meetings held in the CDS office in Porto de Moz. The survey focused on listing the farming activities carried out by families in each community. The comprehensive survey conducted for all communities of Porto de Moz resulted in a list of the main productive activities adopted by the different communities for agriculture, ranching, forest use, and fishing. Based on this list, buffalo ranching, logging, and fishing were identified as the most representative local practices.

In a second research phase, eight communities with longer-term experience with these practices were selected as case studies. Case studies were conducted in four communities of buffalo ranchers, in two communities that use traditional logging systems, and in two communities that maintain fishing agreements. These communities were visited in two periods of the research: (1) July 2010 to August 2013, to monitor and describe practices, and (2) November to December 2021, to evaluate the permanence of the initiatives. Both authors had previous contact with local communities since the first author is originally from the neighbor municipality of Senador José Porfírio and did part of his studies on community forestry in Porto de Moz from 2006 to 2008, and the second author was born in Porto de Moz, still lives in the Resex Verde para Sempre, and did his masters on local management systems in Porto de Moz from 2013 to 2015 [16,25]. The case studies field research was conducted by both authors and interviews with community members were carried out in Portuguese for all the cases.

The case studies focused on the characterization of the production systems by monitoring the practices adopted by the families. All practices of a complete production cycle (for example, from the identification of trees up to timber commercialization) were monitored. The fieldwork included the description of practices and photographic records of each production step. This study summarizes the essential practices that characterize each of the assessed production systems and the main technical solutions developed by local communities.

The case study protocol was based on joining the local families in their farming practices to understand how they performed their activities. For example, for characterizing the traditional forest management system, the research team joined the farmers in their activities to select the trees to be felled, process the logs, transport the timber, etc. For such, the visits were planned for the moments when activities were taking place. In all cases, the research team stayed at the community villages for the time required to follow the complete production cycle (from a couple of days to one week, depending on the case).

The respondents were the community members performing the activities. In the case of buffalo ranchers, respondents included the family members responsible for the herds' management in the field and also the members that produced the cheese at home. In the case of forest management, respondents were the extractors performing the activities in the forest but also the members of the group responsible for the commercialization and processing of the timber. For the fishing agreements, respondents included community members that took part in the elaboration of the agreements and also the ones that help communities to enforce the agreements.

## 4. Results

### 4.1. Buffalo Ranching

#### 4.1.1. Coverage

Buffalo ranching has become one of the most important activities for the rural population of Porto de Moz in the last four decades. Families that rear buffalo are found in 65 of the 107 (61%) local and rural communities of the municipality (Figure 1). Most buffalo ranchers are based in the lowland area between the Xingu and Amazon Rivers, the north region of the municipality of Porto de Moz, since they raise buffalo in floodplain areas where grass grows naturally. The herd varies from 10 to 100 animals per family in most communities, with few ranchers having more than 100 buffalos.

Buffalos are reared for meat and milk. The buffalo is a source of cash income for families when sold for slaughterhouses and the milk is used for cheese production as a source of daily basis income in addition to fishing in floodplain areas and to agriculture and logging in dryland areas.

### 4.1.2. Ranching System

Buffalo ranching is carried out by extended families that live close to each other. The animals are traditionally reared in open fields with natural native grass and there are no fences dividing the lands (Figure 2a). The boundaries between one rancher and another are marked by natural limits, such as trees or streams. The family relationships in this system help the agreements on the use of common spaces.

Ranchers use two main management systems, according to the natural conditions. Those that are only in floodplain areas (have no dryland in their farms) manage the herds in stilt corrals with free access to native grass areas. Those in transition zones between floodplain and dryland areas manage their cattle in natural fields in floodplain areas when the water is lower and lead the cattle to planted grazing fields in drylands in flood seasons when natural grazing fields are covered by water. In both cases, the ranching depends on the availability of natural pastures and the rusticity of the animals.

In the ranching system in floodplain areas, families live in stilt houses, and the animals are kept in stilt corrals in the rainy season. In this system, the main challenge is to ensure that there is enough feed for the cattle in flood periods, mainly for calves that cannot swim large distances to reach grazing fields. The main technical alternatives developed by ranchers for better management, considering local environmental conditions, are as follows:

- Rotation between floodplain areas—management of natural pastures with alternation between different areas with natural grass over the year. In most cases, this practice implies the building of two or more stilt corrals to allow the rotation between different areas of native pasture in floodplain areas;
- Supplying grass in stilt corrals—families that have only floodplain areas cut grass in fields for calves and female buffalos that stay in the stilt corral during and after the pregnancy period (Figure 2b);
- Preparation of feeding areas alongside the stilt corrals—some families started to build paddocks next to stilt corrals as reserved grazing fields for flood seasons. These areas serve as a source of grass for young animals in the stilt corral or easier grazing because of their close location (Figure 2c).

In transition areas between floodplain and dryland areas, the animals remain for approximately six months in each environment. The herd stays in the floodplain area during the drought period (August to January), since the grazing in natural fields results in high milk yield and high levels of fat in the milk. The predominant native grass species in floodplain areas are rabo-de-rato, pomunga, and perimembeca (*Hymenachne amplexicaulis*, *Leersia hexandra*, and *Paspalum repens*, respectively). In the flood season (February to July), the herd is taken to dryland areas, which are commonly planted with braquiarão or quicuio grasses (*Brachiaria brizantha* and *Brachiaria humidícola*, respectively) (Figure 2d). When the ranchers have no grazing field in drylands, they commonly rent pasture areas for the whole herd.

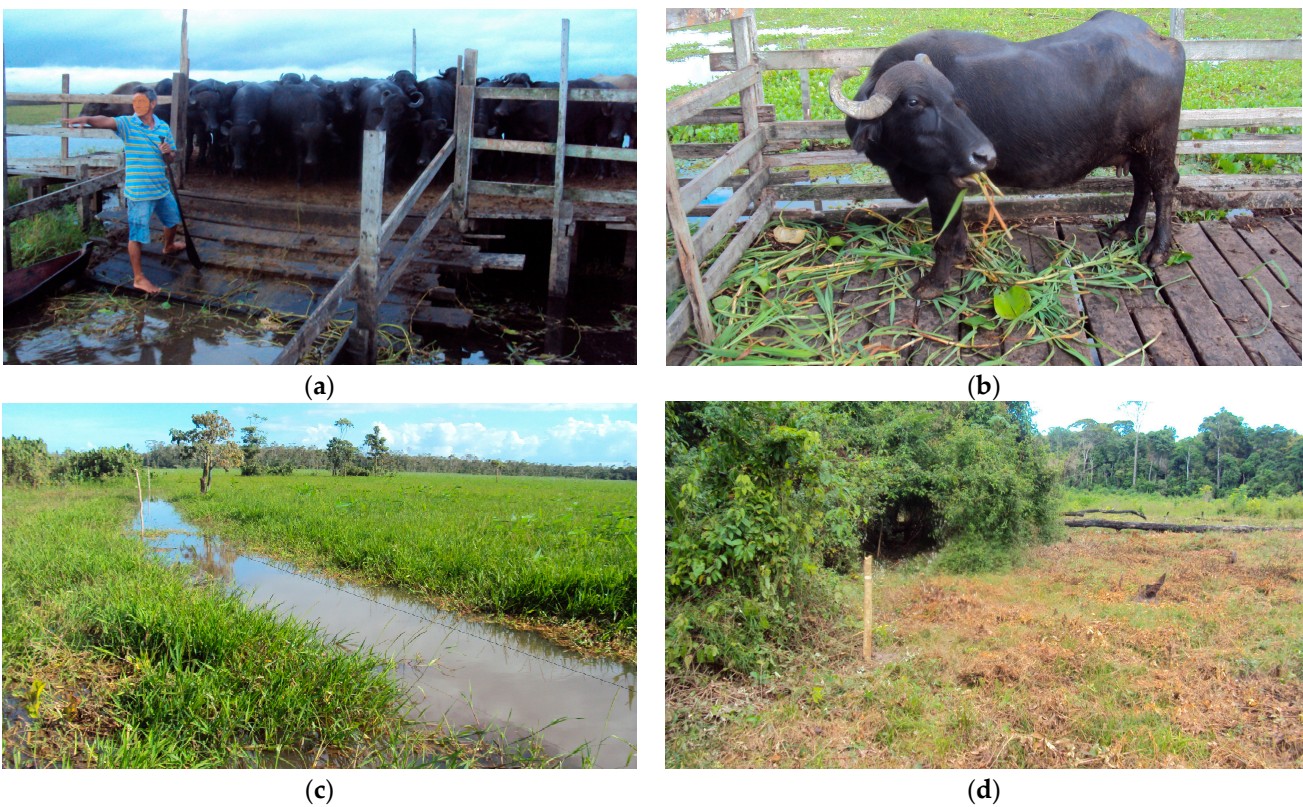

**Figure 2.** Main practices for buffalo ranching used in floodplain areas in Porto de Moz. (**a**) Stilt corral. In floodplain areas, buffaloes are reared in open areas during the day and sleep in stilt corrals during the night. (**b**) Pregnant cow in a stilt corral. Cows and their calves are kept in stilt corral in the pre and post-delivery during the flood season. (**c**) Rest area with electrical fence. These areas are either surrounded by wood boards or electrical fences. Some families keep the electrical fences in the flood season by lifting the wires according to the water level. (**d**) Grazing field planted in dryland. One of the main challenges for farmers is the long-term maintenance of grass productivity since the reform of pastures are expensive.

### 4.1.3. Governmental Action

The Resex Verde para Sempre included areas with buffalos under traditional ranching systems, which raised the need for a dialogue between local families and government agencies. This effort for dialogue was intended to ensure the continuity of buffalo ranching in the Resex area since the legislation of the Brazilian National System of Conservation Units (SNUC—Law 9.985/2000) does not permit the rearing of large-size livestock such as cattle in conservation units.

The uncertainties felt by families about the possibility to keep their activities and the suspension of any type of support to buffalo ranching by the government in the Resex area, including governmental credits (Pronaf), are leading to disruptions of ranching systems. Although most of the Resex rural communities have families rearing large-size animals (either buffalo or cattle), the Ministry of Environment considers that there is an incompatibility between bovine ranching and the SNUC legislation. In the case of Resex Verde para Sempre, the ranchers searched for solutions, mobilizing the communities to promote their practices.

### 4.2. Traditional Logging
### 4.2.1. Coverage

Forest products are an essential part of the local economy for the riparian populations in the Amazon, where they are used for local construction and as a source of income when it is commercialized. Out of the 107 rural communities of Porto de Moz, 64 communities

(60%) have families working on logging (Figure 3). However, only the community of Arimum, which received external financial and technical support, managed to have a formal plan for community forest management approved by the government.

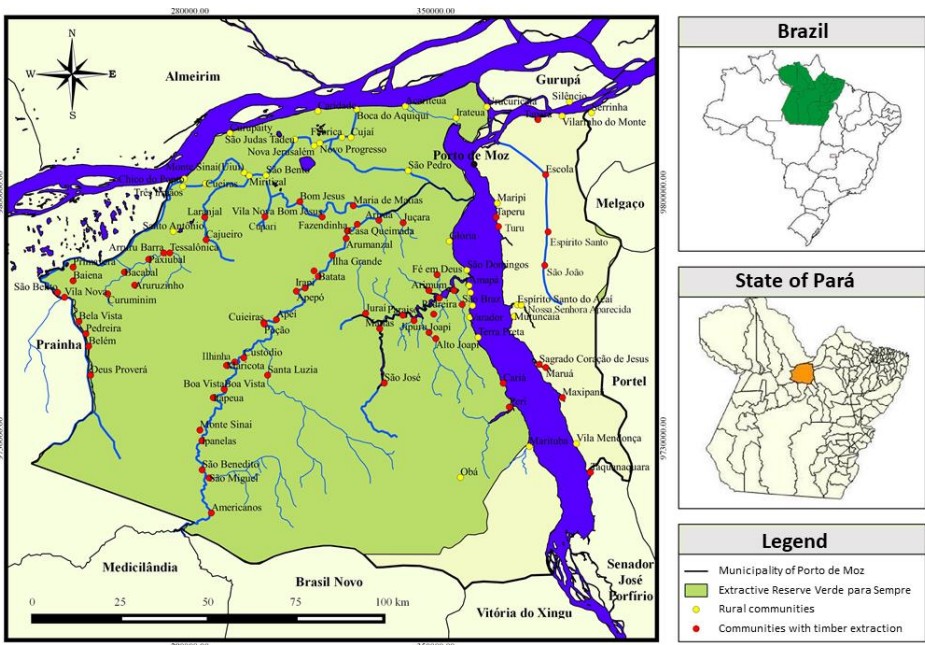

**Figure 3.** Communities where families log their forests in the municipality of Porto de Moz. Source: Fieldwork.

The local logging system is based on kinship between family groups. Therefore, the most common form of organization for logging activity is based on members of one family or family groups, and not the community as a whole (including all families in one community). The organization by family groups is explained by relationships of trust between family members.

### 4.2.2. Production System

Logging methods and log processing practices vary among communities. However, there are characteristics in common such as the choice of areas to be explored, the selection of trees to be extracted, the practices of tree felling, processing, and transporting, and the marketing strategies.

### Definition of Logging Areas

The area to be explored is chosen based on the availability of the desired species, the absence of conflicts with nearby dwellers, and the distance to transportation venues (natural streams or roads opened by logging companies). Logging is practiced mainly in three types of areas:

- Private areas—The family explores their individual area according to their needs, workforce, and available species. The exploration of these often small areas reduces as the stocks of the main commercial species are depleted and families tend to look for other areas;
- Community areas—More than 10 communities demarcated community areas to resist invasions by loggers before the Resex was created. Logging occurs in communal areas at the rear of private areas, which are often established along the river margins. The use of these community areas is based on norms pre-established by members of a community that, in principle, do not accept the entry of outsiders for logging. In general, these areas have high timber stocks and are reserved for future use;

- Free access areas—These areas are usually in the headwaters of streams or roads abandoned by logging companies, away from community centers and individual properties. They are commonly explored by specific family groups, sometimes from different communities, although it is common to have families from the same community working together. In many cases, specific areas are explored by specific family groups and can be considered areas of common use. However, nothing prevents other groups from also having access to these areas.

Selection of Trees

Depending on the quantity desired, the surveying (inventory) of trees is carried out as follows: (a) during hunts, trees are found and mapped mentally, having a stream or an open pathway as a reference; (b) information on areas with incidence of some species is provided by travelers such as middleman trading goods to extractors; and (c) conventionally, with a forest expert from the community searching the forest for species of interest in remote areas that are not claimed by other communities.

Felling

The trees of interest are commonly felled using three basic criteria:

- Tree size—Circumference greater than 200 cm in height and straight trunk length higher than 5 m, to preserve young individuals;
- Logistics of transport—Short distance between the tree location and the stream or river margin, depending on the transport mode available;
- Workforce—Extractivists that work by themselves have difficulty placing trees greater than 400 cm circumference for sawing; therefore, they are not preferred for felling.

The felling is carried out using chainsaws (Figure 4a). The worker performs a hollow test before the felling by the friction of a machete or ax in the tree trunk and analysis of the emitted sound. Some workers also perform the test by drilling the trunk with the tip of the chainsaw up to the tree core. Trees that present signs of dirt or drilling without powder are often hollow inside and are inappropriate for felling; thus, they are preserved. When the trunk is not hollow, the tree is felled towards the direction of the heavier branches.

Processing (Sawing)

The wood is sawn in the ground the trees are felled, using only chainsaws. The tree is sawn in logs according to the desired lengths. The logs are then marked with a string soaked in burnt oil (dark-color oil used as an automotive lubricant). The first sawing divides the log into two parts (bands), which are positioned with the sawing side upwards. The wood is marked with a string again in the lateral for the first sawing to remove the sap-wood from both sides of the band. This procedure removes parts that undergo a second cut with the sawing side downwards (Figure 4b).

Transport

The transport is conducted using wheels, animal traction (commonly buffalos), micro-tractors, old trucks, or manually, carrying the sawn wood on the shoulders or shifting the logs with a log drag system. Families that have old trucks or microtractors fell trees from up to approximately 8 km of the river, which serve as means of transportation. Families that use wheels (Figure 4c) or animal traction fell trees from up to approximately 3 km of the river. This distance tends to be shorter for manual transportation.

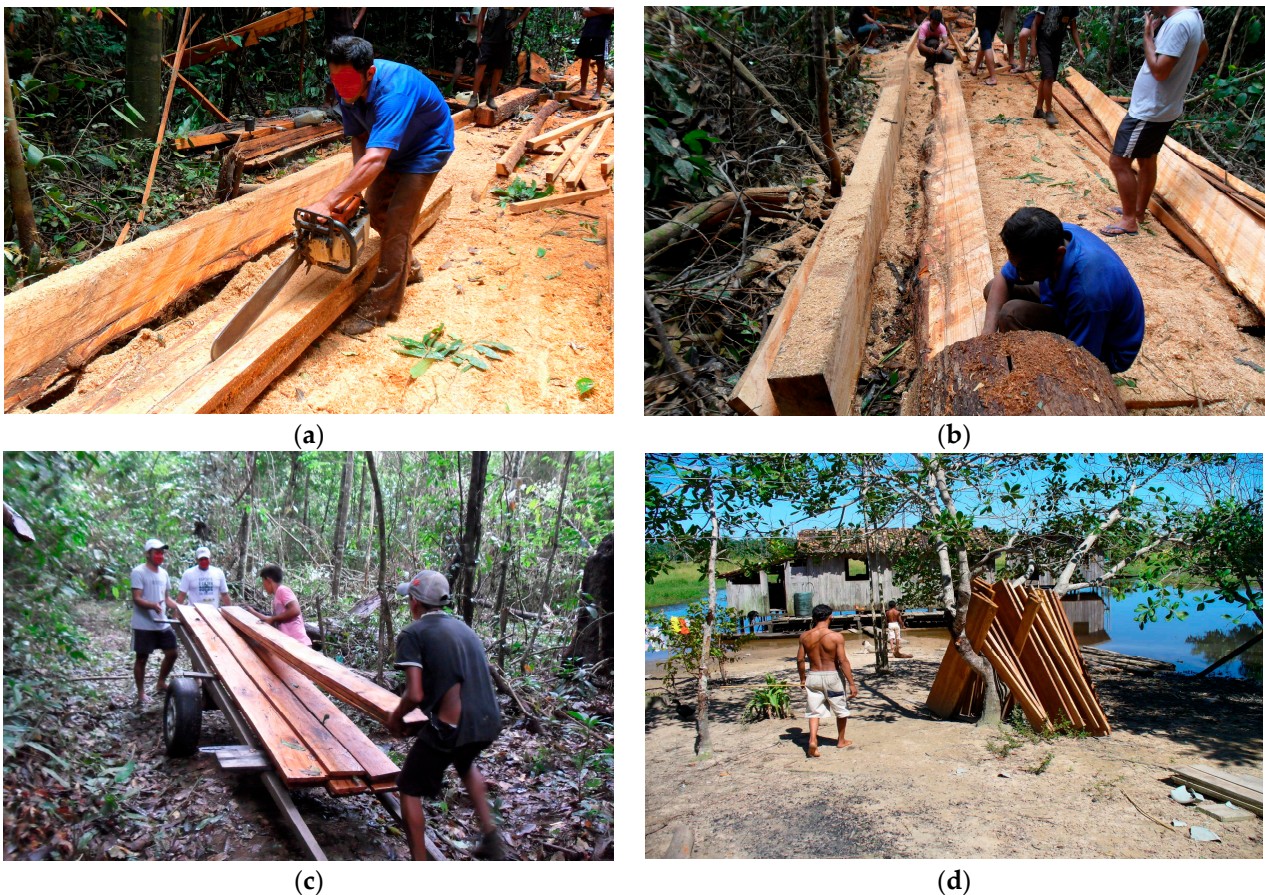

**Figure 4.** Main practices for timber extraction used by families in Porto de Moz. (**a**) Felling of trees with a chainsaw after the area and trees of interest are selected. (**b**) Timber saw by extractive workers using a chainsaw. (**c**) Wheels used by families to transport sawn timber through the forest up to the river margin. (**d**) The wood acquired by families is used for constructions of houses, boats, and stilt corrals, and for trade in local markets.

Marketing

The local demand for timber creates an important market for families, which is added to the demand of markets in Porto de Moz and other neighbor municipalities (Figure 4d). Piquiá (*Caryocar villosum*) is the most commonly used species for the structure of boats built both locally and in neighbor municipalities. Muiracatiara (*Astronium lecointei*), angelimpedra (*Hymenolobium petraeum*), angelim-rajado (*Marmaroxylon racemosum*), louro-faia (*Euplassa pinnata*), cedro-cheiroso (*Cedrela odorata*), and marupá (*Simarouba amara*) are used for furniture such as wardrobes, tables, and chairs. Hardwood (timber with high density) such as maçaranduba (*Manilkara huberi*), cumaru (*Dipteyx odorata*), ipê (*Handroanthus serratifolius* and *Handroanthus impetiginosus*), jatobá (*Hymenaea courbaril*), and angelim-vermelho (*Dinizia excelsa*) is extracted mainly to be sold to Belém, the largest city in the state of Pará. On the other hand, softwoods, such as jabutirana (*Erisma uncinatum*), quaruba (*Vochysia paraensis*), marupá (*Simarouba amara*), and cedrorana (*Cedrelinga cataeniformis*), are used for building walls of houses and walls and windows of boats (the superior parts of the boats that have no direct contact with water once the boat is sailing). The main wood marketing systems are:

- Local consumers from the local communities—The timber is mainly demanded for the construction of houses, corrals, stilt corrals, fences, and boats;
- Consumers in local cities—The timber is sold in the city for furniture makers, planer owners for wood processing, and the general public;

- Middlemen—the tradesman sells essential supply products for extractive workers for later payment, which is received in sawn wood;
- Trader (regional market)—Different from the middlemen, the trader provides inputs or purchases the wood with cash payments. The production is then resold for log industries that reprocess and legalize the products by fraud (i.e., the wood is cut illegally and then the trader uses fraud to make it look legal).

### 4.2.3. Governmental Action

The formal demands of the Sustainable Forest Management Plan (SFMP) contrast with traditional practices, interests, and capacities of the family farmers in the Amazon. The SFMP presupposes the adoption of techniques that generate costs with training and acquisition of equipment, endorsement by a forest engineer, creation of a company or cooperative, and a formal management plan to be assessed by a governmental environmental agency.

The promotion of community forest management based on the SFMP disregards that the whole organization system is based on kinship ties between families and not on community models. Since families cannot afford the costs of the SFMP, the government needs to analyze the possibility of recognizing traditional systems for timber extraction and simplify these administrative procedures.

### *4.3. Fishing Agreements*

### 4.3.1. Coverage

Community-based fishing agreements have been established over the years as important collective management strategies for protecting fishing resources in the Amazon [40]. These agreements seek to meet the interests of riparian communities that fish for consumption and for sale, commercial fishermen, and governmental agencies that regulate the activity. In Porto de Moz, 22 of the 107 communities (21%) take part in the seven existing fishing agreements (Figure 5).

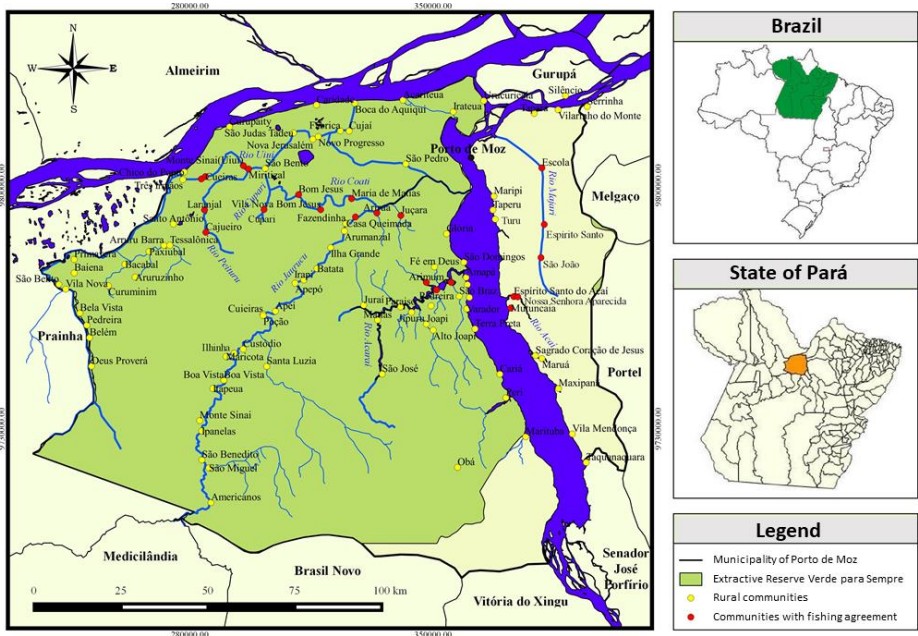

**Figure 5.** Communities with fishing agreements in the municipality of Porto de Moz. Source: Fieldwork.

The fishing agreements include the practices and rules accepted by local groups, which regulate the exploration of fishing resources. The agreements do not necessarily need to be written or regulated by governmental agencies. Unwritten informal agreements can have good results when there is no risk of invasion by fishermen from other places, and

when the quantity of caught fish is small and does not, in the long term, affect the existing fish stocks.

However, when the fishing carried out by local dwellers or outsiders has commercial purposes, stocks can be depleted quickly. Growth in market demand tends to make fishermen increase their production, generating scarcity of target species and, gradually, breaking traditional agreements for the use of these resources. These cases have required written agreements and formal regulation of the established rules to ensure better control of fishing activities with support from governmental regulatory agencies. Thus, communities that cannot enforce the rules of the agreement by themselves have to have state support.

4.3.2. Management System

The fishing agreements in the studied area were created due to food scarcity caused by predatory fishing with the use of equipment with high capture capacity such as small-mesh nets, by fishing in the spawning period, by the entry of outside fishermen in community fishing areas, and by the capture of fishes with sizes below those allowed for commercial exploitation. The first step used for developing the agreement was to synthesize the problem or problems (Table 1).

**Table 1.** Key technical information used to support the formulation of the fishing agreement of the communities of the Coati and Cupari rivers.

| Main Species of Fish | Fishing Season | Aim |
| --- | --- | --- |
| Tucunaré (*Cichla ocellaris*) and aruanã (*Osteoglossum bicirrhosum*) | July to September | Commercialization and local consumption |
| Pirarucu (*Arapaima gigas*) | August to October and January to February | Commercialization and local consumption |
| Tambaqui (*Colossoma macropomum*) and pirapitinga (*Piaractus brachypomus*) | April to July | Local consumption (mainly) |

When the problem caused tensions between residents or communities, or between residents and outside fishermen, the main stakeholders were identified. The identification of these key actors allowed the communities to find the root and coverage of the problem and solutions. The communities realized who was causing harm, the area covered, and whether this coverage tended to increase over time.

Before starting the definition of norms, the group draws a map of the different natural environments used for fishing by families, identifies the main users of these environments, and then invites them for meetings. Once the main issues are made clear, the norms are drafted considering the neighbor communities.

When the solution found for the problem points to the constitution of an agreement or norms for fishing, they are formulated with clear objectives. The agreements are not the end, but the means to reach one or more objectives. Agreements are developed with caution to avoid favoring some groups and penalizing others. All interests and ideas need to be considered in the agreement to ensure its fulfillment in the long-term.

However, this does not mean that the agreements should contain many norms, on the contrary. For example, agreements formulated by communities of the Coati and Cupari Rivers have three prohibitions: fishing with a gillnet, use of timbó (*Ateleia glazioviana*—a plant that causes paralysis in fishes), and the killing of wild ducks. With these three straightforward norms, communities succeeded in keeping control of extractive activities for more than 40 years. Table 2 shows the situation of fishing agreements in Porto de Moz by 2021.

**Table 2.** General situation of collective management of fishing resources in Porto de Moz.

| Rivers | Communities | Agreement | Situation | Advances |
|---|---|---|---|---|
| Coati and Cupari | Maria de Matias, Vila Nova Bom Jesus, Vila Bom Jesus, and São João | Community fishing agreement | Agreement no longer fulfilled by local families, fishing with the use of gillnets, fishing of pirarucu fish in the prohibited period, position of attorney of ICMBio contrary to the endorsement of the agreement. | The ICMBio did not endorse the fishing agreement, local norms were removed resulting in predatory fishing practices. |
| Acaí | Espírito Santo, Nossa Senhora Aparecida, and Santa Ana do Mutuncaia | Community fishing agreement | Communities control access to fishing resources by norms established since 1993. | Agreement acknowledged by the Para State Secretary of Fishing and Aquaculture by a Normative Instruction. |
| Jaurucu | Juçara, Carmelino, and Ariruá | Community fishing agreement | Moderate fishing in higher production periods, mainly for subsistence. | Agreement sent to the ICMBio, pending decision. |
| Uiui | Cuieiras, Monte Sinai, and Santa Luzia | Community fishing agreement | Control of fishing of acari fish established, the communities respect periods for commercial capture, and use equipment as established in the norms. | Twelve years of agreements with results in maintaining fish population, mainly acari; agreement written and sent to IBAMA, which returned no decision. |
| Majari | Espírito Santo, São João, and Escola | Use plan for the PAEX | Fragile agreement with non-compliance by part of local families that fish commercially. | Succeeded meetings for deciding agreement norms, but the implementation is still a challenge. |
| Acaraí | Por Ti Meu Deus, Pedreira, and Arimum | Coexistence agreement | Created in 1996 for coexistence with external fishermen on the use of water and forest resources. The agreement has been revised following assessments of its effectiveness and new norms were created. | Successful maintenance of a stable exploration of resources without conflicts and predatory fishing by outsiders. |
| Peituru | Miritizal, Laranjal, and Cajueiro | Fishing agreement | Local fishermen are fulfilling the agreement; results are already seen in increases in fish abundance. However, outside fishermen have been invading the area of the agreement. | Communities successfully agreed on general norms to regulate fishing and keep the agreement based on dialogues between external and local fishermen. |

PAEX = State Extractive Settlement Project; ICMBio = Chico Mendes Institute for Conservation of Biodiversity; IBAMA = Brazilian Institute of Environment and Renewable Natural Resources.

### 4.3.3. Governmental Action

The acknowledgement of local norms by environmental agencies is essential to ensure the fulfillment of fishing agreements, especially by external fishermen and members of the riparian communities. This official recognition assists in the definition of norms consistent with the current legislation, strengthens the agreements, and protect the community members managing conflicts.

All fishing agreements in Brazil have been regulated by administrative acts as normative instructions (instruções normativas) and ordinances (portarias) (IBAMA Normative Instruction nº 29/2002). The agreements are published by the agency responsible for the management of the area in which the communities are based. For the case of conservation units such as extractive reserves, the agency in charge is the Chico Mendes Institute for Conservation of Biodiversity (ICMBio). The main demand posed by communities is that

governmental agencies ratify the existing local agreements as a means to strengthen the norms by having them formalized.

In Porto de Moz, the Pará State Secretary of Fishing and Aquaculture (Sepaq) ratified the Acaí River agreement, strengthening the communities' organization and helping them to promote a good management of resources. However, the ICMBio did not accept the Coati River and Cupari River agreement, advocating for a management plan for the whole conservation unit. As reported by interviewed community members, the main reasons given by the ICMBio were that government cannot hand over to communities the duty of defining norms and that the local agreement lacked scientific evidence endorsing its efficacy. As a result, fishing resources were left unprotected, and communities were penalized due overfishing.

## 5. Discussion

The productive inclusion of local populations is one of the main challenges for the sustainable development of the Amazon. Although there is evidence of the local communities' capacity to manage natural resources [16,22,24], governmental support is required to legitimize local production systems [2]. The development of production systems adapted to the region can make local small farmers and traditional communities succeed and resist to the advance of the agricultural frontier over their areas by outside loggers and large commodity farmers [6,18].

This study highlights the role of traditional practices in sustainable development based on a cross-comparison of national regulatory frameworks with the local systems for the management of natural resources. Such an approach is important to showcase how national frameworks can be aligned to the social reality. Table 3 summarizes the main challenges for the consolidation of the local production systems described in this study.

**Table 3.** Cross-comparison of national regulatory frameworks with the local management systems.

| Production System | Regional Coverage | Governmental Action | Ruling Legislation | Main Challenges |
|---|---|---|---|---|
| Buffalo ranching in floodplain areas | Found in 61% of the assessed communities. Present in the downstream of the Amazon River (including the Marajo Island), in its confluence with the Xingu River, and in other natural pasture areas. | Prohibition of rearing large animals such as buffalo in conservation units. | National System of Conservation Units (SNUC—Law 9.985 from 2000) only allows small-size livestock in conservation unities [41]. | Recognition of the traditional activity carried out before the creation of the conservation unit. |
| Traditional timber extraction | Found in 60% of the assessed communities and in most of the Amazon. | Lack of recognition of local practices, and promotion of formal forest management. | National Forestry Code (Law 12.651 from 2012) requires formal forestry management plans for timber extraction [42]. | Reframing forest management by incorporating local practices. |
| Fishing agreements | Found in 21% of the assessed communities. Most riparian communities in the Amazon fish and some have fishing agreements. | Governmental control of fishing, often not recognizing local fishing agreements. | IBAMA Normative Instruction nº 29 (from 2002) defines fishing agreements that can be formalized by the responsible agency (ICMBio in conservation unities) [43]. | Acknowledgement of the fishing agreements by the government since governmental support helps the maintenance and strengthening of the agreements. |

SNUC = Brazilian National System of Conservation Units.

*5.1. Benefits of Local Farming Systems Vis-à-Vis Exogenous Initiatives*

Part of the sustainable development initiatives by the Brazilian government and NGOs—many of them, such as the Amazon Fund, with foreign support—have a modernizing approach, which implies a strong control of the communities' way of life [44]. This top-down approach is no longer consistent with the paradigm of post-productivist rural development [1], which values local and traditional know-how [2] and farmer-driven innovations [3].

Governments tend to hesitate in recognizing local small-scale systems mainly because regulating agencies face challenges for controlling free-riding by large-scale farmers, loggers, and fishermen that could claim to be commercializing products from small-scale areas. However, countries in South America have already proven their capacity to enforce policies that clearly differentiate small from large-scale farmers such as in the case of Pronaf credit to family farmers in Brazil and the certification for traditional dairy products in different countries [45]. For example, Bolivia enacted changes in legislation and policies to increase the control of the forestry sector by rural communities [46].

Externally driven initiatives promoted for local communities and family farmers in the Amazon are often restricted to demonstrative pilot areas subsidized by foreign donors [9]. While one of the most comprehensive efforts to list ongoing development initiatives identified 148 cases in the whole Brazilian Amazon [15], this study, conducted in one single municipality, identified 151 existing initiatives taking into account only the thee studied systems (65 communities managing buffalo, 64 communities managing timber, and 22 communities adopting fishing agreements). Since the studied municipality represents only around 0.34% of the area of the Brazilian Amazon, it is possible to expect a huge number of promising local initiatives to be found once we zoom-in and include in the sample endogenous management systems and farmer-driven innovations.

As a consequence, despite being deemed as powerful forces of change in the Amazon, externally driven initiatives that succeed in transforming communities are often insufficient to advance sustainable development to a broader societal scale [15]. The focus on external market niches is a particular challenge since local markets are disregarded, harming the local society that relies on affordable goods for their livelihoods. It can be the case of the current bio-economy paradigm focused on high-value external market niches [47]. In contrast, the buffalo meat, fish, and timber products reported in this study are essential for food, housing, and transport systems (boats) by the local communities.

Often neglected by exogenous initiatives, family farmers in the Amazon are developing local management systems based on existing capacities and information on natural resources. The cases of buffalo, forest, and fishing management systems have a common characteristic since all are examples of technologies and management practices developed by farmers using locally available tools. Local technologies have a high potential to be transformed into development alternatives adapted to the local conditions.

Since these local systems are virtually everywhere and vary according to local conditions, they should not be seen as models to be replicated to other areas but rather as solutions to build upon. What is missing is a development approach building on the potential of these different local systems.

*5.2. Acknowledgement of Local Systems*

The main challenge communities face is promoting the acknowledgement of their local systems in order that they can evolve into sustainable practices for local development. This study shows a conflict of legitimacy between local communities and external stakeholders since it is not clear whether the communities themselves or governmental agencies have the legitimacy to conceive and to propose the development pathways for the Amazon, especially for communities that are in conservation areas [33,48]. In many cases, these uncertainties and unbalanced forces cause losses [34,35] and result in conflicts and frustrations for communities, compromising sustainable local development [36,49].

The need for assertive action is urgent since rural exodus and the disruption of local production systems can undermine later efforts. In the case of traditional buffalo ranching in Porto de Moz, for example, the suspension of the rural credit after the creation of the Resex left families without conditions for the investments needed to endure the seasonal floods in the region.

The socioeconomic inclusion of family farmers should begin with the acknowledgement of existing local practices. The best example of the dysfunctionality of the externally driven projects for the Amazon is the community forest management model, which is found in a few pilot experiences that are often financially inviable [9]. The alternative for forest governance presented by communities has greater potential for success, and government should reframe its forest management concepts to support local initiatives.

Without governmental recognition, local management systems are fragile, mainly when facing external threats, such as invasion by outside loggers and large farmers [14]. For example, in the case of fishing agreements in Porto de Moz, the acknowledgment of the Acaí River agreement by the Pará State Secretary of Fishing and Aquaculture (Sepaq) strengthened the communities' organization and promoted good management of the resources. However, when the ICMBio did not recognize the Coati River and Cupari Rivers agreement, the fishing resources were left unprotected since communities had no governmental support for limiting access by external fishermen to their areas. In this case, the communities themselves ended up punished since fishing stocks were depleted by overfishing practices by both local and external fishermen. In other words, an imperfect agreement is still better than no agreement at all, and the government's failure to acknowledge and support existing agreements is doing more harm than good.

This study showed the existence of representative local production systems developed based on local capacities that require governmental recognition for their consolidation, as seen elsewhere [33,38]. The government can play an important role by legitimizing these local systems and strengthening them [22].

The potential for local farming and governance systems to form the basis for sustainable development in the Amazon was also emphasized in other studies [20,21]. This study reveals that this debate continues to be timely and shows case that local systems with the potential to underpin sustainable development are virtually being developed by every community in the Amazon. Despite their huge potential, local systems remain neglected by most of the governments, donors, and NGOs who continue prioritizing top-down externally driven development initiatives.

### 5.3. Potentials and Limits of Local Farming System

Regarding the sustainability of these local systems, particularly in ecological terms, this study described conservation-related practices such as the rotation of grazing grounds for buffalo ranching; the selection of specific trees for timber extraction, and the local fishing norms that may favor the conservation of fish natural stocks. Given their limited impact and adaptation to local environmental conditions, these local practices have the potential to be transformed into sustainable management systems.

However, the existence of local farming practices does not guarantee sustainable management systems. For example, if the population of buffalo exceeds the carrying capacity of natural fields, it can jeopardize sustainable development based on farmer-driven management practices. The environmental performance of these local systems can be assessed and improved if they are properly supported by external stakeholders, such as in the case of fish management in the Mamirauá Reserve and many other examples [50].

Regarding sustainability on the social and economic sides, their endogenous nature indicates that local systems may suit the capacities and needs of Amazonian populations. The fact that communities continue to perform and adapt these local practices is an indicator of their financial viability. Local systems fit within cultural norms and social structures (e.g., family connections), labor and labor relations, low cost of local technologies, local ecological knowledge, ease to repair equipment, etc.

## 6. Conclusions

This study revealed the existence of local farming systems developed by family farmers over decades of experience that can potentially underpin the sustainable development of the Amazon. Local buffalo, forest, and fishing management systems were found in 61%, 60%, and 21% of the 107 studied communities, respectively. The large coverage of these local systems contrasts with pilot initiatives of exogenous production systems that are limited to a small number of communities in pilot projects.

The studied local farming systems are essentially endogenous, with technical solutions developed based on the existing conditions, although incorporating the available technologies. Local farming systems are essential for the livelihoods of small farmers and are often the main source of affordable goods, such as meat, timber, and fish for local communities.

These results help to bring back the debate over top-down vs. bottom-up management approaches and to draw scholarly and policy attention to local populations and their resource use systems in search for alternative options for sustainable development. For such, attention should be given to farmer-led technological innovation, local governance, and recognition of marginalized knowledge.

The consolidation and improvement of local systems require governmental recognition and support of local agricultural, livestock, and extractive initiatives. The lack of governmental acknowledgment weakens these local systems and demobilizes communities, undermining the conservation of natural resources. Without governmental support, local management systems are fragile, mainly when facing external threats, such as invasion by external large farmers, loggers, and commercial fishermen.

These lessons are fundamental for governments, NGOs, and donor agencies that should reconsider their focus on externally driven, top-down concepts such as community forestry, payments for environmental services, and bio-economy-related eco-business initiatives targeting external niche markets as opposed to regional markets that benefit local families with affordable goods. Instead, donors should help communities to improve their existing systems, which can benefit from external support for upgrades in technological, economic, marketing, environmental, and social aspects.

**Author Contributions:** Conceptualization, G.d.S.M. and C.W.S.B.; Methodology, G.d.S.M. and C.W.S.B.; Validation, G.d.S.M. and C.W.S.B.; Formal Analysis, G.d.S.M.; Investigation, G.d.S.M. and C.W.S.B.; Resources, G.d.S.M. and C.W.S.B.; Data Curation, G.d.S.M. and C.W.S.B.; Writing—Original Draft Preparation, G.d.S.M.; Writing—Review and Editing, G.d.S.M.; Project Administration, G.d.S.M. and C.W.S.B.; Funding Acquisition, G.d.S.M. and C.W.S.B. All authors have read and agreed to the published version of the manuscript.

**Funding:** This research was funded by the European Commission in Brazil through the project "Governança de Recursos Naturais por Pequenos Produtores da Amazônia". Funding number: DCI-NSAPVD/2010/208-221.

**Institutional Review Board Statement:** Exempted from Institutional Review Board Statement based on the RESOLUTION No. 510 of the Brazilian government that states that: "Article 1. Single paragraph. Will not be registered or evaluated by the CEP/CONEP system: I—Public opinion poll with unidentified participants".

**Informed Consent Statement:** Informed consent was obtained from all subjects involved in the study.

**Data Availability Statement:** Not applicable.

**Acknowledgments:** We would like to thank all farmers who collaborated on this study.

**Conflicts of Interest:** The authors declare no conflict of interest.

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
