# Peer review of "The Neglected Solutions: Local Farming Systems for Sustainable Development in the Amazon"

_world, doi:10.3390/world4010011_

Round 1

Reviewer 1 Report

The paper discusses a very important and topical subject, namely the role of traditional practices in sustainable development. The research findings are built on an extensive field work and involve participatory approaches. The main merit and novelty of the paper is the cross-comparison of national regulatory frameworks with the local, traditional norms. Such approach is important to showcase how national frameworks can be aligned to the social reality. In overall, the authors demonstrate an extensive and profound knowledge of the topic, on which I offer my congratulations. 

My suggestions are mostly related to the description of the research methods, in order to prove the robustness of the results. 

Introduction 

I would suggest the restructuring of the introduction part to give more clarity of the objective of the paper. The volume and quality of the information is sufficient; however, a better structure would greatly support the readers to understand the research objective. For example, an introduction formula can be the following: problem statement/rationale, research question and objective, review of existing literature, statement about the novelty and contribution to the science, and the roadmap of paper. Beyond this example, there are many introduction formulas that can help the formulation of a coherent and concise introduction. The authors may wish to select the most appropriate one. 

Materials and methods

I would strongly recommend including a map about the selected 107 communities in the context of the region. 

The paper refers to a comprehensive survey conducted in all communities. Nevertheless, neither the survey template nor the set of questions/data collection protocol is displayed or detailed. I would suggest that the authors include the survey in annex, or the information content in a table format. Reference to this additional information would help understand how the survey was conducted, thus demonstrating the rigor of the scientific approaches. Also, a summary of the characteristics (or summary statistics) of the respondents would be desirable to gain a better insight into the background of the implementation of the survey. 

The above observation applies to the description of case study approach too. A more detailed description of the case study protocol would be appreciated, including the framework, the questions, selection of respondents, etc. 

The paper refers to the previous works of the authors in the communities. This information requires a more rigorous reference to the other works of the authors, by either including a citation of the previous works or summarizing the findings of the previous works. It would be crucially important to explain which results of the previous works are used in this paper. 

Results

One of the most important merits of this paper is the comparison of national regulatory framework and the social reality in the study areas. This is clearly a very important argument to explain how the evolution of the national policy and legal frameworks ignored the traditional practices. It is also interesting to see the implication of the national frameworks on the livelihoods. To better emphasize this part, I would suggest that the authors include a summary table of the policies and legislation, against which the traditional practices are assessed. Such summary table could include the title of the policy/legislation, the relevant objective, the administrative tier of implementation, and its contribution to sustainable development. 

The sub-sections on “Governmental action” would benefit from a more structured and systematic write-up, for example phased as the following: names of relevant national policies/legislative documents, part of traditional practices compliant with the document, part of traditional practices non-compliant with the document. 

Discussion

Some traditional practices are considered as not sustainable in the “Results” section, yet, the “Discussion” attributes limited adverse impact of the traditional practices. Please reconcile the arguments in the two sections. 

References

Please include the title of national policies/legislative documents in the list of references

Author Response

Dear reviewer, thank you for your kind and constructive comments. We benefited from the fact that you summarized the paper so well by adding your summary in the discussion section. We restructured the introduction and split it into an introduction and a theoretical framework for the sake of clarity. We also provided more details in the methods section to better explain the survey and case studies. We included in Table 3 the policies and legislation, against which the traditional practices are assessed and also cited them in the references. Also, the discussion section was revised to clarify the limits of this study in terms of sustainability assessment. Also for the other comments, we did our best to address them accordingly whenever it was possible. Thank you for helping us to better frame the core of this study.

Reviewer 2 Report

Dear authors. In the attachment you will find my observations and recommendations. Good luck!

Author Response

Dear reviewer, thank you for your kind and constructive comments. As for the structure of the introduction, we have now split the introduction section into an introduction and a theoretical framework for the sake of clarity, so that lines 69-77 are now at the bottom of the introduction as you recommended. We also did the recommended editing. As for the references, in case the manuscript is accepted, we will contact the journal to check for all the changes to be made and have them all at once, including your recommendations. Also for the other comments, we did our best to address them accordingly whenever it was possible. Thank you for helping us to better frame the core of this study.

Reviewer 3 Report

The paper revisited the top-down initiatives and introduced endogenous initiatives on farming (local and traditional practices, like ranching, logging, and fishing) for sustainable development in Amazon. There are still some issues with the manuscript.

1. The introduction section. At the end of the section, it is better to summarize the research content and innovation of this paper to give a preface in one sentence or paragraph to the overall research.

2. The materials and methods section. It is better to add an introduction to the research area (including the climate, terrace, economy, society, etc.) to give enough background, although the paper briefly introduced the urban-rural background in the introduction section.

3. Figures 1, 2, and 3 could be merged into one figure to be the distribution of various types of communities. It is only for reference.

4. The description of the local farming practices (ranching, logging, and fishing) would not enough reflect sustainable development. It needs more background information, like the changes in population, and land scrab. If the population or Buffalo exceeds the carrying capacity, it’s difficult to explore sustainable development on farmer-driven management, but more give up sustainability to survive.

5. The discussion section is better written in several paragraphs, including the advantages/sustainability of the local farming system (to outcome the unrealistic of exogenous initiatives), suggestions for sustainable development in Amazon, and implications to other regions (or limits of local farming system).  

Author Response

Dear reviewer, thank you for your kind and constructive comments. As for the structure of the introduction, we have now split the introduction section into an introduction and a theoretical framework for the sake of clarity. We also included some more details on the research area in the methods. The discussion section was revised to clarify the limits of this study in terms of sustainability assessment. We also split the discussion into different subsections following your last comment. Also for the other comments, we did our best to address them accordingly whenever it was possible. Thank you for helping us to better frame the core of this study.

Round 2

Reviewer 3 Report

It would be best for the authors to respond to the comments one by one. If they didn't revise the manuscript, please give a reasonable explanation.

Author Response

Dear reviewer, thank you for your kind and constructive comments. As for the structure of the introduction, we have now split the introduction section into an introduction and a theoretical framework for the sake of clarity. We also included some more details on the research area in the methods. The discussion section was revised to clarify the limits of this study in terms of sustainability assessment. We also split the discussion into different subsections following your last comment. Specifically:

  1. The introduction section. We have now split the introduction section into an introduction and a theoretical framework for the sake of clarity. The specific objectives summarize the structure of the manuscript and the innovations are presented in the conclusion section.

  1. The materials and methods section. We included more details on the research area in the methods. We consider that details such as climate, terrace, economy, society, etc are easily available at the internet for the study area.

  1. Figures 1, 2, and 3 could be merged into one figure. It is only for reference. We preferred to keep them as they are since it allows for better visualization of the communities practicing each system.

  1. The description of the local farming practices (ranching, logging, and fishing) would not enough reflect sustainable development. We added a new subsection at the bottom of the discussion section to address your suggestion. Thank you for emphasizing that.

  1.  The discussion section is better written in several paragraphs. We added the subsections in the discussion section following your recommendation.

Round 3

Reviewer 3 Report

accept